# Neutrophil Extracellular Traps in Pediatric Inflammatory Bowel Disease: A Potential Role in Ulcerative Colitis

**DOI:** 10.3390/ijms252011126

**Published:** 2024-10-16

**Authors:** Rachel Shukrun, Victoria Fidel, Szilvia Baron, Noga Unger, Yoav Ben-Shahar, Shlomi Cohen, Ronit Elhasid, Anat Yerushalmy-Feler

**Affiliations:** 1Pediatric Hemato-Oncology Research Laboratory, Tel Aviv Medical Center, Tel Aviv 6423906, Israel; shukrun.rachel@gmail.com (R.S.); vicufidel@gmail.com (V.F.); baron_szilvia@yahoo.com (S.B.); 2Department of Pediatric Hemato-Oncology, “Dana-Dwek” Children’s Hospital, Tel Aviv Sourasky Medical Center, Tel Aviv 6423906, Israel; 3Faculty of Medical and Health Sciences, Tel Aviv University, Tel Aviv 6997801, Israel; unger.noga@gmail.com (N.U.); shlomico@tlvmc.gov.il (S.C.); anatyf@tlvmc.gov.il (A.Y.-F.); 4Department of Pediatric Surgery, “Dana-Dwek” Children’s Hospital, Tel Aviv Sourasky Medical Center, Tel Aviv 6423906, Israel; yoav_bs@hotmail.com; 5Pediatric Gastroenterology Institute, “Dana-Dwek” Children’s Hospital, Tel Aviv Sourasky Medical Center, Tel Aviv 6423906, Israel

**Keywords:** inflammatory bowel disease, neutrophil, neutrophil extracellular traps (NETs), ulcerative colitis, Crohn’s disease, prognostic factor

## Abstract

Inflammatory bowel disease (IBD), encompassing Crohn’s disease (CD) and ulcerative colitis (UC), is a chronic inflammatory condition of the gut affecting both adults and children. Neutrophil extracellular traps (NETs) are structures released by activated neutrophils, potentially contributing to tissue damage in various diseases. This study aimed to explore the presence and role of NETs in pediatric IBD. We compared intestinal biopsies and peripheral blood from 20 pediatric IBD patients (UC and CD) to controls. Biopsy staining and techniques for neutrophil activation were used to assess neutrophil infiltration and NET formation. We also measured the enzymatic activity of key NET proteins and evaluated NET formation in UC patients in remission. Both UC and CD biopsies showed significantly higher levels of neutrophils and NETs compared to controls (*p* < 0.01), with UC exhibiting the strongest association. Peripheral blood neutrophils from UC patients at diagnosis displayed increased NET formation compared to controls and CD patients. Interestingly, NET formation normalized in UC patients following remission-inducing treatment. This pilot study suggests a potential role for NETs in pediatric IBD, particularly UC. These findings warrant further investigation into the mechanisms of NET involvement and the potential for targeting NET formation as a therapeutic strategy.

## 1. Introduction

Inflammatory bowel disease (IBD) encompasses a group of chronic inflammatory conditions of the gastrointestinal tract, primarily categorized into Crohn’s disease (CD) and ulcerative colitis (UC) [1]. In pediatric populations, IBD presents unique challenges, with more aggressive and extensive forms than adult-onset IBD [2]. The etiology of IBD is multifactorial, involving genetic predisposition, environmental factors, and alterations in the gut microbiome [3]. The chronic inflammatory response in IBD is associated with increased intestinal epithelial permeability, which promotes the massive recruitment of neutrophils [4]. Neutrophils contribute to IBD pathogenesis through a variety of mechanisms [5,6].

Neutrophil extracellular traps (NETs) are web-like structures released by activated neutrophils, composed of DNA, histones, and various antimicrobial proteins [7]. Initially identified for their role in trapping and killing pathogens, NETs have since been implicated in a variety of inflammatory and autoimmune diseases [8]. While NETs play a crucial role in host defense, their dysregulation can contribute to tissue damage and sustained inflammation [9].

The involvement of NETs has garnered significant attention among adults with IBD. Studies have shown that elevated levels of NETs in the mucosa of IBD patients have been associated with the promotion of tissue damage [10]. Interestingly, the role of NETs appears to differ between UC and CD. Studies have correlated NETs mainly with inflammation in UC, with high expression of proteins associated with NETs being observed in bowel samples from patients with UC [6,8]. Importantly, NETs were also shown to be generated by neutrophils in the peripheral blood of adult patients with active IBD [11,12,13].

The role of NETs remains relatively unexplored in pediatric IBD. Our group pioneered the investigation of NETs in pediatric IBD samples, demonstrating their presence for the first time [14]. However, large knowledge gaps regarding their specific contributions to disease processes still exist. This study aimed to address this by investigating NET formation in pediatric IBD, with a particular focus on potential differences between patients with UC and CD.

## 2. Results

### 2.1. Elevated Neutrophils and NETs in Pediatric IBD Tissue Samples Compared to Controls

Twenty patients with IBD were included: 11 with CD and 9 with UC (50% males). Their median age at diagnosis was 15.0 (4.2–17.7) years. The demographic and clinical characteristics of the study cohort are detailed in Table 1. We first investigated the presence of neutrophils and NET formation in intestinal tissue samples from pediatric IBD patients compared to controls (N = 9 (55% females)). Their median age at diagnosis was 15.8 (10–16.9) years. A significantly higher number of infiltrating neutrophils was observed in both CD (328.9 ± 53.91) and UC (425.6 ± 72.98) tissues compared to controls (88.33 ± 21.76) (Figure 1A, *p* = 0.0003). Additionally, increased NET formation was observed in both CD (34.77% ± 3.91) and UC (39.57% ± 4.21) tissues compared to controls (8.88% ± 3.04), (Figure 1B, *p* < 0.0001). Representative images further illustrate these findings (Figure 1C).

### 2.2. Neutrophil Infiltration and NET Formation Were Higher in Inflamed vs. Non-Inflamed Intestinal Tissue

To further evaluate the association between inflammation and NET formation, we analyzed non-inflamed intestinal tissue samples from the same IBD patients. We compared these samples to inflamed intestinal tissue biopsies. This analysis revealed significantly higher levels of infiltrating neutrophils in inflamed intestinal tissue (425.6 ± 72.98 UC and 328.9 ± 53.91 CD), compared to the non-inflamed tissue (13.4 5 ± 4.32 UC, 178 ± 29.69 CD, *p* = 0.0112 and *p* = 0.0319, respectively), (Figure 2A,B). In addition, significantly higher NET formation was observed in inflamed intestinal tissue (39.57% ± 4.21 UC and 34.77% ± 3.91 CD), compared to the non-inflamed tissue (13.45% ± 4.32 UC and 11.01% ± 2.84 CD, *p* = 0.0006 and *p* = 0.0001, respectively) (Figure 2C,D). Representative images again support these findings (Figure 3).

Given the limited data on NET formation in peripheral blood of pediatric IBD patients and after observing their presence in intestinal biopsies (Figure 1), we investigated NET formation in neutrophils isolated from blood. The control group included healthy pediatric patients undergoing elective surgery. Neutrophils isolated from patients with UC at diagnosis displayed significantly elevated NET formation upon stimulation with PMA compared to both healthy controls and patients with CD (59.58% ± 5.48, 34.26% ± 2.59, and 34.14% ± 3.74 for UC, PC, and CD, *p* = 0.012 and *p* = 0.186, respectively) (Figure 4A). Representative images further illustrate this finding (Figure 4D). To explore the potential mechanisms underlying these differences, we additionally measured the enzymatic activity of core NET proteins, neutrophil elastase (NE), and myeloperoxidase (MPO), in IBD patients at diagnosis. However, no significant changes in enzymatic activities were observed compared to controls (Figure 4B,C). Next, to further validate our findings, we conducted additional experiments using a novel antibody developed by Tilley et al. [15]. This antibody specifically targets a histone H3 cleavage event that is characteristic of NETs, providing a more direct and specific assessment of NET formation. Our results using this new antibody were consistent with our previous findings, further supporting the presence of increased NETs formation in peripheral blood neutrophils from UC patients (Appendix A).

### 2.3. Peripheral Blood NET Formation as a Potential Biomarker for Disease Activity in Pediatric UC

Having observed elevated NET formation in UC patients at diagnosis, we next investigated the impact of immunomodulatory treatment on this process. We analyzed NET formation in peripheral blood samples obtained from patients with UC in remission, defined by a PUCAI < 10 (Figure 5A). Importantly, NET formation in these patients with UC in remission decreased significantly and was comparable to that observed in the healthy pediatric control group (*p* < 0.05, *p* = 0.962, respectively, Figure 5A). Representative images further illustrate this finding (Figure 5D).

We additionally evaluated the enzymatic activity of NET core proteins, NE and MPO (Figure 5B,C). While a decrease in the enzymatic activity of both NE and MPO was observed in patients with UC in remission compared to those at diagnosis, this decrease was not statistically significant (*p* = 0.504 and *p* = 0.1239 for NE and MPO, respectively).

## 3. Discussion

This study investigated the role of NETs in pediatric IBD, with a specific focus on ulcerative colitis. Our data shows that both UC and CD tissues displayed significantly higher levels of infiltrating neutrophils and NET formation compared to healthy controls. Notably, patients with UC exhibited the most pronounced increase. Moreover, neutrophils isolated from patients with UC displayed a significantly greater capacity for NET formation compared to controls and patients with CD. This confirms and expands on data from studies showing an increased abundance of NET-related proteins in adult patients with UC [11,12,13].

The observed UC-specific increase in NET formation compared to CD hints at potential disease-specific mechanisms at play. However, this finding is not without controversy. Previous studies in adults present conflicting results, with some finding elevated NETs in both UC and CD [13,16,17,18] while others report no significant NET increase in CD [8,19]. These discrepancies likely stem from variations in experimental methods and patient cohorts. Additionally, several mechanisms might contribute to the UC-specific increase in NET formation.

One explanation challenges the reliance on peptidylarginine deiminase 4 (PAD4) as the sole marker for NET formation. While Dinallo et al. [8] ruled out increased NETs in CD biopsies based on the absence of elevated PAD4 expression, recent research suggests that alternative PAD4-independent pathways exist [20]. Another potential mechanism hinges on the activity of key enzymes like MPO and NE involved in NET formation [21]. While our study has not demonstrated significant differences in their overall activity levels between UC and CD, their specific regulation and localization within the NETs could differ. UC-specific neutrophil activation might lead to altered enzyme activity or localization within the NET’s structure, impacting its overall function. Despite potential overlap in IBD mechanisms between age groups, significant differences in pediatric NET formation likely exist due to factors like immune system development, gut microbiome, disease characteristics, treatment approaches, and response to therapy [22]. Thus, further pediatric-focused research is crucial to understanding the UC-specific increase in NETs observed in this study.

There is an ongoing debate regarding the optimal method for identifying NETs in tissue samples [20]. While traditional approaches have relied on the co-localization of NE and citrullinated histone H3 (cH3), other studies have highlighted the limitations of this method. As previously mentioned, not all NET formation pathways involve citrullination. Recent studies have demonstrated that NETs can form through PAD4-independent pathways, which do not involve citrullination. For instance, Tsourouktsoglou et al. (2020) investigated the role of histones, DNA, and citrullination in NET-induced inflammation. Their findings suggest that Cl-amidine, a potent inhibitor of PADs, effectively blocked histone citrullination but failed to inhibit NET formation in human neutrophils, indicating that NET formation can occur through alternative pathways [23]. Our study employed a comprehensive approach that included the use of an H3 antibody capable of recognizing both citrullinated and non-citrullinated forms of H3. This allowed us to capture both citrullination-dependent and citrullination-independent NET formation pathways, providing a more accurate and comprehensive understanding of NET involvement in pediatric IBD.

Despite advancements in diagnosis and treatment, the lack of reliable biomarkers to accurately assess disease severity and reliably indicate remission remains a significant challenge in managing IBD [24]. Current laboratory markers, like C-reactive protein (CRP) and fecal calprotectin, offer limited specificity and can be influenced by factors unrelated to disease activity [25]. This hinders optimal treatment decisions and patient monitoring. The data presented in this study on NET formation in IBD offers a promising avenue for the development of a novel biomarker. Our findings demonstrate a clear association between NET formation and intestinal inflammation in UC. Furthermore, the normalization of NET formation upon achieving remission in UC patients suggests a potential correlation with disease activity. While further investigation is warranted, the ability to quantify NET formation in readily accessible samples, such as peripheral blood, could provide a valuable tool for clinicians. A non-invasive and specific biomarker reflecting disease activity would be a significant advancement in IBD management. Future studies should explore the development of standardized assays to measure NET formation in IBD patients and assess its potential as a biomarker for disease severity and monitoring treatment response.

The detrimental role of NETs in IBD, particularly UC, underscores their potential as therapeutic targets. This notion is further bolstered by the observation that several effective IBD medications, including infliximab and mesalamine, exhibit NET-suppressive properties [26,27,28]. Exploring the mechanisms behind these effects and optimizing their NET-reducing capacity could lead to significant advancements in IBD treatment. Furthermore, research into novel medications specifically targeting NET formation or degradation pathways holds immense promise. Strategies that either inhibit NET production or enhance NET degradation mechanisms could offer entirely new avenues for IBD management.

This study has limitations. The sample size was relatively small, and further research with larger cohorts is needed for definitive conclusions. Additionally, the mechanisms underlying the observed differences between UC and CD require further investigation.

In conclusion, this study demonstrates elevated neutrophil infiltration and NET formation in pediatric IBD tissues, with a stronger association observed in UC. NET formation appears to be linked to active inflammation and might be modulated by immunomodulatory treatment. Our findings highlight the potential role of NETs in pediatric IBD pathogenesis, particularly in UC, and suggest that compounds interfering with NET release could be useful for dampening the detrimental immune response in UC. This study lays the groundwork for future research to investigate the functional significance of NETs in pediatric IBD, their potential as biomarkers, and their therapeutic possibilities.

## 4. Materials and Methods

### 4.1. Study Population

This is a single-center prospective study conducted at Dana Children’s Hospital of Tel Aviv Medical Center, Tel Aviv, Israel. All parents signed an informed consent form in accordance with the Declaration of Helsinki, Institutional Review Board, Tel Aviv Sourasky Medical Center, Tel Aviv, Israel 0502–19-TLV.

Children and adolescents (≤18 years of age) diagnosed with CD and UC according to the revised Porto criteria [29] at the Pediatric Gastroenterology Institute, “Dana-Dwek” Children’s Hospital, Tel Aviv Sourasky Medical Center, were included. As controls, patients with functional gastrointestinal disorders diagnosed according to ROME IV criteria [30] that had a normal upper and lower gastrointestinal endoscopy and histology were selected. Healthy pediatric patients undergoing elective surgery served as controls.

### 4.2. Study Design and Data Collection

We collected prospectively demographic and clinical data on IBD characteristics. Disease location, behavior, and extent were defined according to the Paris classification [31]. Disease activity at diagnosis of IBD and at follow-up was measured using the weighed Pediatric Crohn’s Disease Activity Index (PCDAI) [32] or the Pediatric Ulcerative Colitis Activity Index (PUCAI) [19]. The clinical course, medical therapy, hospitalizations, disease exacerbations, surgeries, laboratory results, and imaging and endoscopic findings were collected.

Mucosal biopsies were taken during endoscopy of children from the study group from the normal and inflamed areas in the terminal ileum and colon. For the controls, mucosal biopsies were taken from macroscopically and microscopically unaffected colonic segments. Peripheral blood samples were taken from children with active IBD, patients with UC in remission, and healthy pediatric patients undergoing elective surgery.

### 4.3. Materials

Phosphate-buffered saline was obtained from Sartorius AG (Göttingen, Germany). Ethylene diamine tetra-acetic acid (EDTA), human albumin, bovine serum albumin (BSA), phorbol 12-myristate 13-acetate (PMA), and Triton X-100 were purchased from Sigma-Aldrich (St. Louis, MO, USA). Poly-l-lysine solution (0.01%) and 4% formaldehyde solution were acquired from Merck (Rahway, NJ, USA).

### 4.4. Isolation of Neutrophils

Human peripheral blood samples (3–9 mL) in EDTA-coated vacutainer tubes (Greiner Bio-One, Kremsmünster, Austria) were obtained from IBD patients and controls. Neutrophils were isolated using the EasySep Direct Human Neutrophil isolation kit (StemCell Technologies Inc., Vancouver, BC, Canada) by immunomagnetic negative selection according to the manufacturer’s instructions. The number of isolated neutrophils was quantified using a Beckman Coulter DxH800 hematology analyzer (Beckman Coulter Inc., Brea, CA, USA), and the final concentration was adjusted to 10^7^/mL in RPMI without phenol red.

### 4.5. Neutrophil Elastase Enzymatic Activity

Enzymatic activity was measured as was previously described [33]. Briefly, 10^5^ neutrophils were lysed and then incubated with chromogenic peptide elastase substrate at a final concentration of 0.5 mM (Calbiochem, Darmstadt, Germany) for 90 min at 37 °C and measured using an iMark Microplate Absorbance Reader (Bio-Rad Laboratories, Inc., Hercules, CA, USA) at 415 nm. NE activity was calculated for 10^6^ neutrophils.

### 4.6. Myeloperoxidase Enzymatic Activity

Myeloperoxidase enzymatic activity was measured as described previously [33]. Briefly, 10^5^ neutrophils were lysed and then incubated with O-phenylenediamine at a final concentration of 50 µg/mL (Sigma-Aldrich) and H_2_O_2_ at a final concentration of 1 mM (Sigma-Aldrich) for 20 min at RT and measured using an iMark Microplate Absorbance Reader (Bio-Rad Laboratories, Inc.) at 450 nm. MPO activity was calculated for 10^6^ neutrophils.

### 4.7. Induction of NETs by PMA

A total of 2 × 10^5^ neutrophils were seeded on coverslips coated with poly-l-lysine and activated by a final concentration of 100 nM phorbol 12-myristate 13-acetate (PMA in DMSO, Sigma-Aldrich) for 3 h at 37 °C and then fixed with a 4% formaldehyde solution.

### 4.8. Immunofluorescent Staining and NET Quantification of Isolated Neutrophils

Following activation with PMA, neutrophils were stained for visualization of NET formation. Cells were incubated with Sytox Green (Invitrogen, Thermo Fisher Scientific, Waltham, MA, USA) and Hoechst 33342 (Sigma-Aldrich) nuclear dyes according to the manufacturer’s instructions. Briefly, this allows for differentiation between condensed and decondensed chromatin. Images were captured using an LSM700 Laser Scanning Confocal Fluorescence Microscope (Zeiss, Oberkochen, Germany). For each sample, three regions of interest containing 100–200 cells were analyzed. NET formation was quantified manually based on the established criteria. Neutrophils without NETs were defined as cells with compact DNA stained by Hoechst 33342. NET-forming neutrophils were defined as cells with diffused DNA stained by Sytox Green. The percentage of NET formation was calculated as the ratio of NET-positive neutrophils to the total number of neutrophils (NET-positive and NET-negative).

### 4.9. Immunofluorescent Staining of Tissue Samples

Paraffin-embedded sections (4-µm thick) of IBD tissue samples were deparaffinized using xylene 100% and rehydrated with ethanol 100% and 70%, performing two changes in each solution for 5 min each. Subsequently, antigens were retrieved by Target Retrieval Solution pH = 6 10 mM citrate 10× (Dako, Glostrop, Denmark) for 30 min at 50 °C in order to protect the NETs from structural deformation [29]. The sections were then left in the respective buffer to cool to room temperature (RT), rinsed with phosphate-buffered solution (PBS), and circled using PAP-pen after drying. Subsequently, the sections were permeabilized for 1 min with 0.5% Triton X-100 in Tris-buffered saline (TBS) at RT, followed by one rinsing step with PBS, and then treated with 10% (5% BSA plus 5% human albumin, 0.1% Tween-20) blocking buffer in PBS for 1h to prevent non-specific binding. Primary antibodies used were rabbit anti-human neutrophil elastase (NE; EMD Millipore 481001, 1:1000, EMD Millipore, Burlington, MA, USA) and sheep anti-human histone 3 (H3; Abcam128012, 1:500, Abcam, Cambridge, UK) in PBS buffer containing 1% blocking buffer and incubated overnight at 4 °C. After washing with PBS twice, secondary antibodies conjugated to fluorescent dyes, goat anti-rabbit Alexa Fluor 488 (Abcam150081, 1:500) and donkey anti-sheep Alexa Fluor 647 (Abcam150179, 1:500) were diluted in PBS buffer containing 1% blocking solution and incubated for 1 h at RT in the dark. The slides were then washed twice with PBS and incubated with DAPI nuclear dye (Invitrogen, Thermo Fisher Scientific, Waltham, MA, USA) for 10 min.

### 4.10. Microscopy and Image Analysis

Imaging was performed on an LSM700 Laser Scanning Confocal Fluorescence Microscope (Zeiss, Oberkochen, Germany) using a 20× objective. Five regions of interest were collected in each section, and image analysis was performed using Image J software (Version 1.x, National Institutes of Health, Bethesda, MD, USA). Neutrophils not forming NETs were defined as those exhibiting a high-intensity signal with NE (green) but a low intensity signal with H3 (red). NET-forming neutrophils were defined as those exhibiting high-intensity NE (green) and H3 (red) signals, showing co-localization (yellow). The percentage of NETs was calculated as the ratio of NET-forming neutrophils to the total number of neutrophils (NET-forming and non-forming neutrophils).

### 4.11. Statistical Analysis

Statistical analysis was performed using GraphPad Prism version 8 (GraphPad Software Inc., San Diego, CA, USA). The data are presented as means ± standard error of the means (SEM). Statistical differences were determined by employing an ANOVA test with Tukey’s multiple comparison post hoc test for comparisons among multiple groups and a two-tailed Student’s *t*-test for comparisons between two groups. A *p*-value < 0.05 was considered statistically significant.

## Figures and Tables

**Figure 1 ijms-25-11126-f001:**
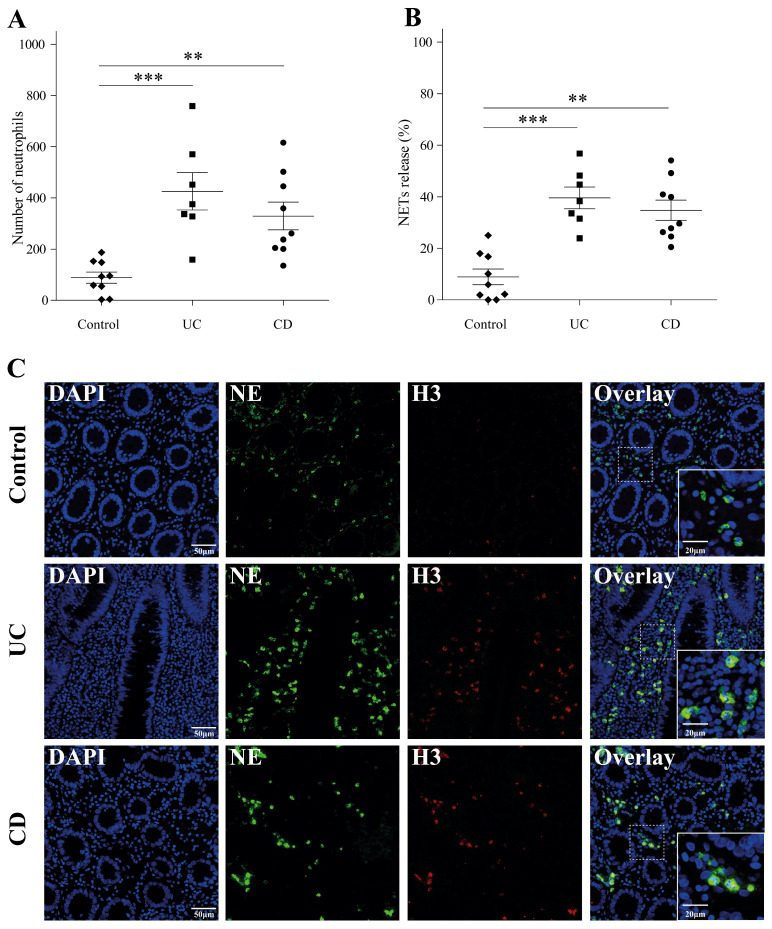
Elevated neutrophils and NETs in pediatric IBD tissue samples. (**A**) The number of neutrophils and (**B**) NET formation was significantly elevated in both UC (N = 9) and CD (N = 11) compared to controls (N = 9) (*** *p* < 0.001, ** *p* < 0.01). (**C**) Representative images of NETs in the tissue samples of patients with UC and CD. The tissues were stained with NE (green) and H3 (red). NET-forming cells are identified by their high H3 signal and NE co-localization (yellow). Scale bar: 50 μm. On the right bottom panel, there is a higher magnification of the selected area (marked by dashed lines), scale bar: 20 μm (right).

**Figure 2 ijms-25-11126-f002:**
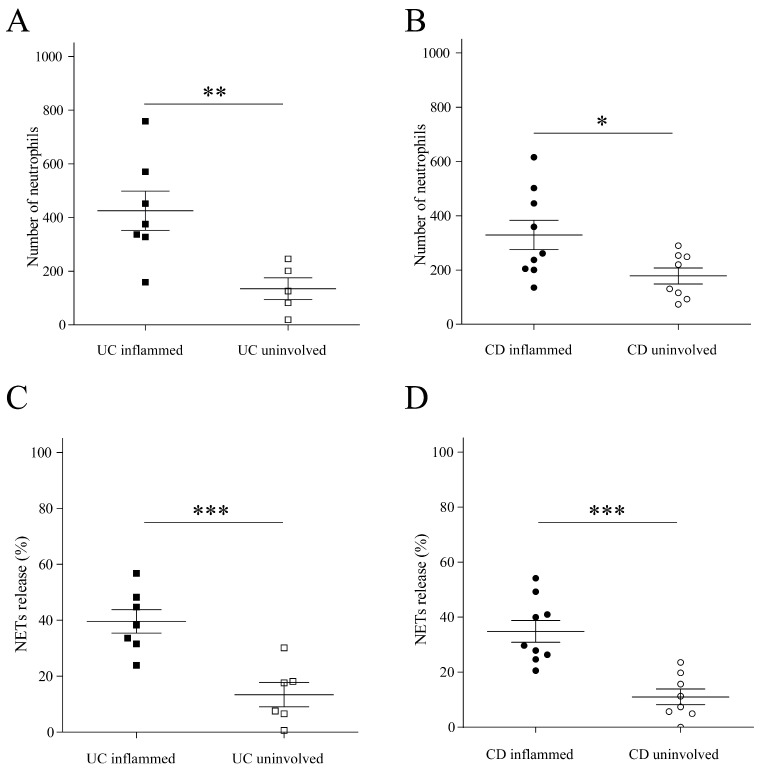
Neutrophil infiltration and NET formation in inflamed vs. non-inflamed intestinal tissue. (**A**,**B**) Significantly higher levels of infiltrating neutrophils were demonstrated in the inflamed intestine of UC (**A**) and CD (**B**) compared to the non-inflamed intestine (* *p* < 0.05, ** *p* < 0.01). (**C**,**D**) Significantly higher levels of NET formation were demonstrated in the inflamed intestine of UC (**C**) and CD (**D**) compared to the non-inflamed tissue (*** *p* < 0.001).

**Figure 3 ijms-25-11126-f003:**
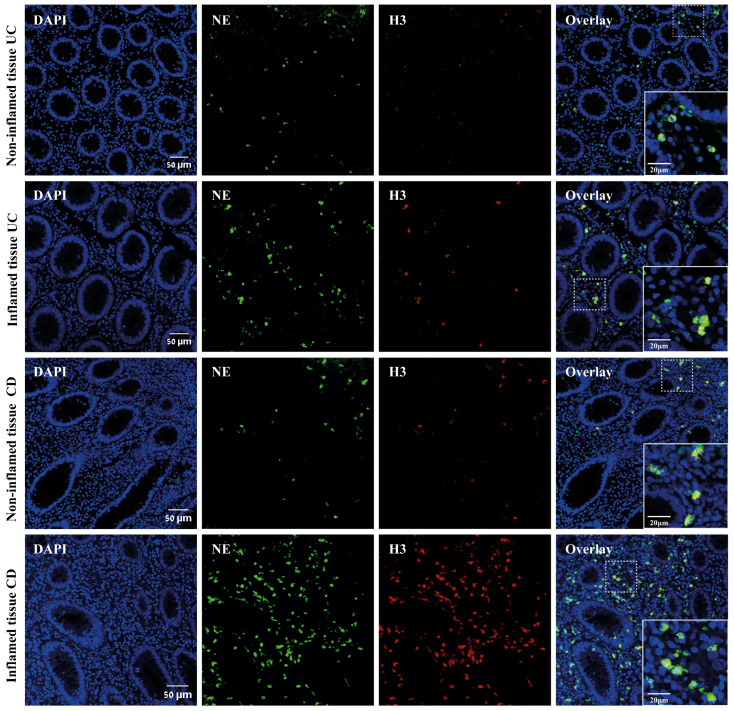
NET formation in inflamed vs. non-inflamed intestinal tissue. Representative images of NETs in the inflamed intestine of UC and CD, compared to the non-inflamed tissue. Tissues were stained with NE (green) and H3 (red). NET-forming cells are identified by their high H3 signal and NE co-localization (yellow). Scale bar: 50 μm. On the right bottom panel, there is a higher magnification of the selected area (marked by dashed lines), scale bar: 20 μm (right).

**Figure 4 ijms-25-11126-f004:**
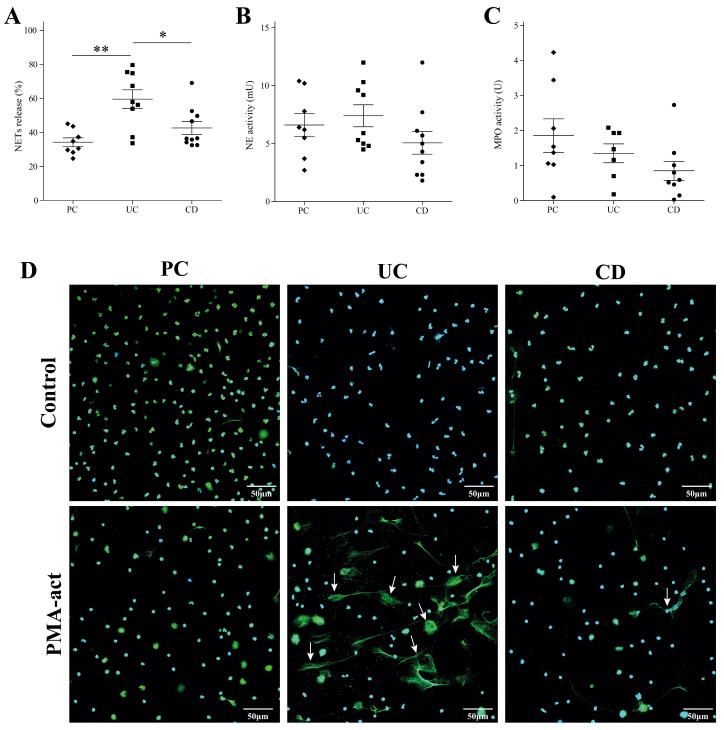
Increased NET formation by neutrophils from pediatric UC patients. (**A**) Neutrophils isolated from patients with UC at diagnosis displayed significantly elevated NET formation upon stimulation with PMA compared to both healthy controls and CD patients (** *p* < 0.01 and * *p* < 0.05, respectively). (**B**,**C**) Enzymatic activity of neutrophil elastase (NE) and myeloperoxidase (MPO) in UC and CD patients at diagnosis showed no significant difference compared to controls. (**D**) Representative images of NET formation in UC at diagnosis. Cells were stained with Sytox Green (green) and Hoechst 33342 (blue) dyes to identify decondensed chromatin. White arrows indicate NET-forming cells. Scale bar: 50 μm.

**Figure 5 ijms-25-11126-f005:**
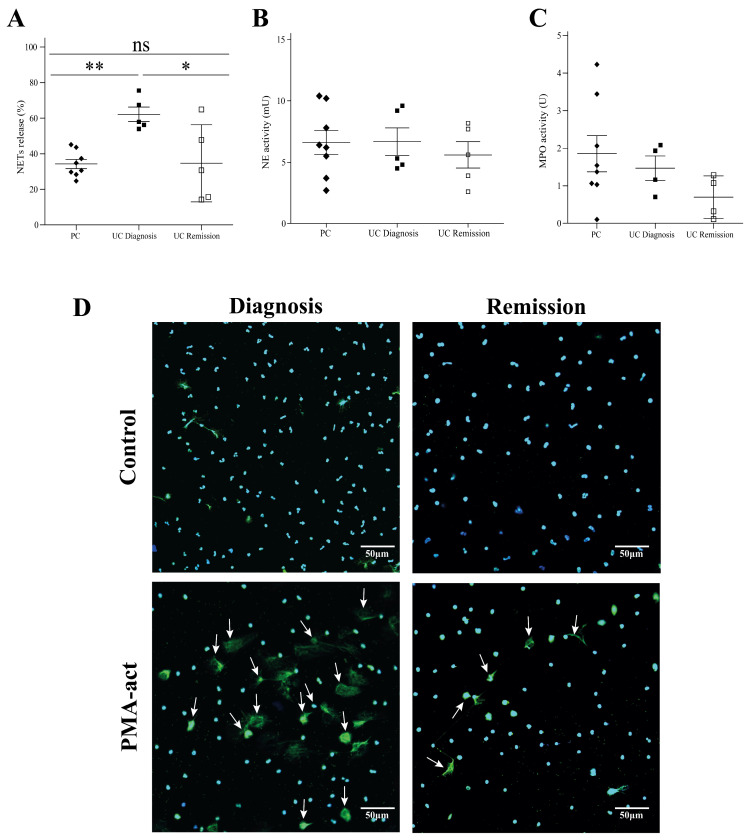
Normalization of NET formation in UC with immunomodulatory therapy. (**A**) Neutrophils from UC patients in remission displayed significantly lower NET formation upon stimulation compared to UC patients at diagnosis (** *p* < 0.01 and * *p* < 0.05, respectively). NET formation in remission was comparable to healthy controls (*p* = 0.962, ns—not significant). (**B**,**C**) Enzymatic activity of neutrophil elastase (NE) and myeloperoxidase (MPO) showed no significant difference between UC patients in remission, at diagnosis, and healthy controls. (**D**) Representative images confirm lower NET formation in UC patients in remission compared to diagnosis. Cells were stained with Sytox Green (green) and Hoechst 33342 (blue) dyes to identify decondensed chromatin. White arrows indicate NET-forming cells. Scale bar: 50 μm.

**Table 1 ijms-25-11126-t001:** Demographic and clinical characteristics of the study cohort.

	All Patients (N = 20)	Crohn’s Disease (N = 11)	Ulcerative Colitis (N = 9)
**Age at diagnosis (years)**	15 (11.5–16)	15 (13–15.2)	15.1 (11.4–16.8)
**Males**	10 (50%)	7 (63.6%)	3 (33.3%)
**Location of CD**			
Ileo-cecal		3 (27.3%)	
Colonic		1 (9.1%)	
Ileo-colonic		7 (63.6%)	
**Behaviour of CD**			
Inflammatory		11 (100%)	
Stricturing/penetrating		0	
Perianal involvement		2 (18.2%)	
**Extent of UC**			
Proctitis			1 (11.1%)
Left-sided colitis			2 (22.2%)
Extensive colitis			1 (11.1%)
Pancolitis			5 (55.6%)
**Severity of UC**			
Never severe			6 (66.7%)
Ever severe			3 (33.3%)
**Extra-intestinal manifestations**	4 (20%)	2 (18.2%)	2 (22.2%)
**PCDAI or PUCAI at diagnosis**		20 (15–34)	40 (35–55)
**IBD severity at diagnosis**			
Mild	9 (45%)	7 (63.6%)	2 (22.2%)
Moderate	8 (40%)	3 (27.3%)	5 (55.6%)
Severe	3 (15%)	1 (9.1%)	2 (22.2%)
**Induction therapy**			
CDED		9 (81.8%)	0
5-ASA		0	8 (88.9%)
Corticosteroids		4 (36.4%)	5 (55.6%)
Biologic agents		10 (90.1%)	4 (44.4%)

CD, Crohn’s disease; UC, ulcerative colitis; IBD, inflammatory bowel disease; PCDAI, pediatric Crohn’s disease activity index; PUCAI, pediatric ulcerative colitis activity index; CDED, Crohn’s disease exclusion diet; 5-ASA, 5-aminosalicylic acid.

## Data Availability

The authors confirm that all the data supporting the findings of this study are available within the article. The raw data are currently not deposited in a publicly available repository. However, we acknowledge the importance of data sharing for scientific transparency and reproducibility. Therefore, the data are available upon reasonable request from the corresponding author (Ronit Elhasid; E-mail: ronite@tlvmc.gov.il).

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
