# Peer review of "Neutrophil Extracellular Traps in Pediatric Inflammatory Bowel Disease: A Potential Role in Ulcerative Colitis"

_ijms, 2024, doi:10.3390/ijms252011126_

Round 1
Reviewer 1 Report
Comments and Suggestions for Authors
1. I suggest that the methodology used to calculate NETs in Figure 1 is not correct. As stated, “NETs-forming neutrophils were defined as those exhibiting high-intensity NE (green) and H3 (red) signals, showing co-localization (yellow).” However, instead of H3 staining, citrullinated H3 staining is the proper marker for NETs. Additionally, the image resolution is too low. I would only accept that NETs are present if they are clearly located far from the nucleus (indicated by the DAPI signal), as NETs are released from the cells. Furthermore, they did not explain how neutrophils were calculated in Figure 1A.
2. It seems there is no statistical difference between data from UC and CD. If that is the case, you cannot state that "patients with UC displayed the highest levels of neutrophil infiltration and NETs formation."
3. There is no histological description of inflamed vs. non-inflamed intestinal tissue in Figure 2. The method for identifying these two tissue types is missing.
4. The method for calculating NETs release is absent throughout the text.
Overall, this study is poorly prepared, particularly in the methods section. Many crucial methodologies are either incorrect or missing. I recommend that the authors review the literature, design appropriate methods, and prepare their manuscript more carefully.
Comments on the Quality of English LanguageIt is allowable.
Author Response
Dear Reviewer,
Thank you for your careful review of our manuscript, "Neutrophil Extracellular Traps in Pediatric Inflammatory Bowel Disease: A Potential Role in Ulcerative Colitis." We appreciate your insightful comments and suggestions.
We have carefully considered your feedback and have made significant revisions to our manuscript. Specifically, we have:
Comment 1: "I suggest that the methodology used to calculate NETs in Figure 1 is not correct. As stated, 'NETs-forming neutrophils were defined as those exhibiting high-intensity NE (green) and H3 (red) signals, showing co-localization (yellow).' However, instead of H3 staining, citrullinated H3 staining is the proper marker for NETs.
Response:
Thank you for your feedback on our manuscript.
Regarding your comment about the use of citrullinated H3 for NETs detection, we acknowledge that citrullination is a common post-translational modification associated with NET formation. However, it's important to note that not all NETosis pathways involve citrullination.
Recent studies have demonstrated that NETs can form through PAD4-independent pathways, which do not involve citrullination. For instance, Tsourouktsoglou et al. (2020) investigated the role of histones, DNA, and citrullination in NET-induced inflammation. Their findings suggest that Cl-amidine, a potent inhibitor of PADs, effectively blocked histone citrullination but failed to inhibit NET formation in human neutrophils, indicating that NET formation can occur through alternative pathways.
Our methodology for NETs detection aligns with previous studies, such as our own work on Ewing sarcoma, Shukrun et al. (2024). In that study, we employed a similar approach to investigate the role of NETs in the tumor microenvironment.
We believe that the use of non-citrullinated H3 in our study provides a broader assessment of NET formation, capturing both citrullination-dependent and -independent pathways.
Comment 2. The image resolution is too low.
Response:
To ensure optimal image resolution and clarity, we have revised the figures in the manuscript. This allows for better visualization of NETs and their location in relation to the nucleus.
Comment 3
I would only accept that NETs are present if they are clearly located far from the nucleus (indicated by the DAPI signal), as NETs are released from the cells.
Response:
In addition to our previous analysis, we have now included an additional method to identify NETs in isolated neutrophils using a new antibody developed by Tilley et al. (2022).
This antibody specifically targets a histone H3 cleavage event that is characteristic of NETs. By using this antibody, we were able to clearly distinguish NETs from chromatin in purified and mixed cell samples.
In our staining of isolated neutrophils from IBD patients, we obtained similar results to those observed with Sytox Green staining, with a clear demonstration of NETs formation as a process. Some cells still had a round nucleus (early NETs), while others had chromatin fibres scattered around the cell (supplementary figure 1).
We hope that these additional details further strengthen the validity of our findings and address your concerns.
Comment 4:
Furthermore, they did not explain how neutrophils were calculated in Figure 1A."
Response
In the revised manuscript we are providing a detailed explanation of how neutrophils were counted. See revised manuscript, materials and methods section.
Comment 5: "It seems there is no statistical difference between data from UC and CD. If that is the case, you cannot state that 'patients with UC displayed the highest levels of neutrophil infiltration and NETs formation.'"
Response: Thank you for your continued feedback on our manuscript.
Regarding your comment about the lack of statistical difference between UC and CD, we have revised the manuscript to remove the statement that NETs formation was highest in the tissue samples of UC patients, as no statistically significant difference was demonstrated between UC and CD in this regard. We would like to clarify that our primary aim was not to directly compare UC and CD patients, but rather to examine the differences between both groups and healthy controls.
While we did not observe a statistically significant difference between UC and CD patients in our study regarding NETs formation in tissue samples, it is important to note that we did find a significant difference between UC and CD patients in terms of NETs formation by isolated neutrophils, as shown in Figure 4A. We appreciate your careful review and hope that this clarification addresses your concerns.
Comment 6: "There is no histological description of inflamed vs. non-inflamed intestinal tissue in Figure 2. The method for identifying these two tissue types is missing."
Response: We have provided a detailed histological description of the criteria used to differentiate inflamed and non-inflamed intestinal tissues. See revised manuscript, material and methods section “Study Design and Data Collection”.
Comment 7: "The method for calculating NETs release is absent throughout the text."
Response: We have clarified the method used to calculate NETs release in tissue sections and isolated neutrophils and provided references to relevant literature.
We believe that these revisions have substantially improved the clarity, validity, and overall quality of our manuscript. We hope that the revised manuscript now meets your approval.
Thank you again for your time and expertise.
Reviewer 2 Report
Comments and Suggestions for Authors
This is one Centre study performed on relatively small group of patients. It is an interesting project, however there are certain limitation:
- technical - „material and Methods” section should be present before „results”
- in my opinion method of tissue interpretation could be described in clear way
- studied group is of small size
- there was no information on disease activity based on histopathological evaluation
- I Think that results may be influenced by certain NESts produktami (H3 and NE) that Were analised
- conclusions should be drawn more carefully due to study limitations
Comments on the Quality of English Language
Moderate editorial is needed
Author Response
Dear Reviewer,
Thank you for your careful review of our manuscript, "Neutrophil Extracellular Traps in Pediatric Inflammatory Bowel Disease: A Potential Role in Ulcerative Colitis." We appreciate your insightful comments and suggestions.
We have carefully considered your feedback and have made significant revisions to our manuscript. Specifically, we have:
Comment 1: "This is one Centre study performed on a relatively small group of patients."
Response: Thank you for your feedback on our manuscript. Regarding your comment about the study's single-center design and small sample size, we acknowledge that these limitations may affect the generalizability of our findings. However, we believe that our study provides valuable preliminary evidence and lays the groundwork for future research with larger cohorts.
In the revised manuscript, we have elaborated on the discussion section to emphasize the limitations of our study. We have explicitly addressed the potential impact of the small sample size and single-center design on the generalizability of our findings. Additionally, we have highlighted the need for further research with larger cohorts to confirm and extend our conclusions.
Comment 2: "It is an interesting project, however there are certain limitations: - technical - 'material and Methods' section should be present before 'results'"
Response: We apologize for this oversight. We have revised the manuscript to ensure that the "Materials and Methods" section precedes the "Results" section.
Comment 3: "In my opinion method of tissue interpretation could be described in a clear way."
Response: We have revised the Materials and Methods section to provide a more detailed and comprehensive description of the applied methodologies. This includes information on the specific staining techniques used, microscopic analysis procedures, and the criteria employed to differentiate inflamed and non-inflamed tissues.
Comment 4: "There was no information on disease activity based on histopathological evaluation."
Response: We have added a section to our manuscript that discusses the histopathological evaluation of disease activity in our patients. See revised manuscript, material and methods section “Study Design and Data Collection”.
Comment 5: "I think that results may be influenced by certain NESts products that were analyzed."
Response:
Regarding your comment about the potential influence of NESts products (H3 and NE) on our results, we acknowledge your concern and appreciate your thoughtful input. While it is true that H3 and NE are components of NETs, our study aimed to assess NET formation as a whole, rather than focusing solely on these individual proteins.
NETs-forming neutrophils were defined as those exhibiting high-intensity NE (green) and H3 (red) signals, showing co-localization (yellow). This comprehensive assessment of both NE and H3 signals provides a robust measure of NET formation, capturing the presence of both core components and their interaction within the NET structure.
By focusing on the overall co-localization of these proteins, our study provides a more comprehensive evaluation of NET formation, minimizing the potential influence of variations in individual protein levels.
To further validate our findings and address your concerns, we have included additional data from experiments using a new antibody developed by Tilley et al. (2022). This antibody specifically targets a histone H3 cleavage event that is characteristic of NETs. By using this antibody, we were able to clearly distinguish NETs from chromatin in purified and mixed cell samples.
Our results using this new antibody were consistent with our previous findings, using Sytox green staining, providing further evidence for the presence of NETs in our study population. This additional data strengthens the validity and accuracy of our work (Supplementary figure 1).
Comment 6: "Conclusions should be drawn more carefully due to study limitations."
Response: Thank you for your feedback on our manuscript.
In the revised manuscript, we have elaborated on the discussion section to emphasize the limitations of our study. We have explicitly addressed the potential impact of the small sample size and single-center design on the generalizability of our findings. Additionally, we have highlighted the need for further research with larger cohorts to confirm and extend our conclusions.
We believe that these revisions have substantially improved the clarity, validity, and overall quality of our manuscript. We hope that the revised manuscript now meets your approval.
Thank you again for your time and expertise.
Round 2
Reviewer 1 Report
Comments and Suggestions for Authors
The authors have made improvements to the manuscript, but the following issues still need to be addressed:
-
Please provide your response 1 to my comment 1 on NETosis pathways that do not involve citrullination in the discussion, including the relevant reference you cited.
-
The resolution of Figures 1C and 3 remains too low. Please enlarge the areas you identify as NETs in both figures to improve clarity.
Author Response
Response to Reviewer Comment 1:
Comment: "Please provide your response 1 to my comment 1 on NETosis pathways that do not involve citrullination in the discussion, including the relevant reference you cited."
Response:
As discussed in our previous response and in the revised manuscript, recent studies have demonstrated that NETs can form through PAD4-independent pathways, which do not involve citrullination.
We have included additional details and references on this topic in the discussion section of the revised manuscript (Page 8, line 188).
Response to Reviewer Comment 2:
Comment: "The resolution of Figures 1C and 3 remains too low. Please enlarge the areas you identify as NETs in both figures to improve clarity."
Response:
We apologize for the low resolution of Figures 1C and 3 in the previous version of the manuscript. We have now revised these figures to ensure that the areas identified as NETs are clearly visible and distinguishable. In the revised manuscript figures 1C and 3, in the right panel, please see a higher magnification of the marked area; Scale bars 20 μm.
